# Images Segmentation Based on Cutting the Graph into Communities

**Sergey V. Belim * and Svetlana Yu. Belim**

Radio Engineering Department, Omsk State Technical University, 644050 Omsk, Russia
* Correspondence: sbelim@mail.ru

**Abstract:** This article considers the problem of image segmentation based on its representation as an undirected weighted graph. Image segmentation is equivalent to partitioning a graph into communities. The image segment corresponds to each community. The growing area algorithm search communities on the graph. The average edge weight in the community is a measure of the separation quality. The correlation radius determines the number of next nearest neighbors connected by edges. Edge weight is a function of the difference between color and geometric coordinates of pixels. The exponential law calculates the weights of an edge in a graph. The computer experiment determines the parameters of the algorithm.

**Keywords:** image segmentation; community; image road segmentation





## 1. Introduction

Segmentation is the division of an image into areas with close pixel color characteristics [1]. Image segmentation is one of the most relevant in computer vision and pattern recognition systems [2–4]. Segmentation uses color intensity to split monochrome images. Segmentation works in the color feature space for color images. Segmentation algorithms depend on the choice of color model. Segmentation results are not strictly defined. The correctness of segmentation algorithms is assessed on the basis of expert assessments. Image collections with test segmentation are used to test segmentation algorithms. Experts perform ground truth for images of these collections [5]. There are no universal segmentation algorithms that work successfully on arbitrary images.

Most segmentation algorithms present the image as values of some function in the nodes of the square grid. Differential based on finite differences are calculated for these functions. Special image points are used for segmentation. This approach has several drawbacks when applied to noisy or illuminated images. Pre-filtering the image reduces the impact of these factors. However, filters blur the image and degrade the quality of segmentation.

Image processing methods based on graph theory have been actively developed recently. A weighted graph is uniquely mapped to an image. Image segmentation is equivalent to partitioning a graph into subgraphs [6–10]. Community allocation on the graph is used in this approach. Finding communities is clustering the graph. Community searches are typically applied to network data. Communities represent a subset of graph nodes that are more strongly connected to each other than to other nodes. The number of edges is a connectivity measure for an unweighted graph. The weight of the edges is considered for the weighted graph. Edge weights between community nodes are greater than with other nodes.

The graph nodes correspond to the image pixels. The graph edges define a measure of pixel similarity. The graph is mapped to the image so that pixels with close color characteristics are strongly related. Defining communities on a graph groups image points by similar color features. The image segment corresponds to the community on the graph.

One of the applied values for segmentation algorithms is understanding road scenes and highlighting the road in the image. This problem is relevant for the inventory of roads and smart vehicles [11]. The task of highlighting a roads segment is difficult due to differences in road types, objects on the roads, and lighting conditions. The main requirements for such an algorithm are the speed of operation and the accuracy of determining road edges. Excess objects on the sides of the road should not fall into the segment.

Algorithms for highlighting communities on graphs provide a high level of image segmentation quality. However, these algorithms have high computational complexity and long running times. These disadvantages do not allow these algorithms to be used directly. Community highlighting algorithms work in the image preprocessing or image postprocessing stages. The purpose of the paper is to develop an image segmentation algorithm based on highlighting communities on a graph with acceptable running time. The developed algorithm is used to highlight the road segment. The algorithm calculates the real road width based on the image segment and optical system parameters.

## 2. Related Works

Communities can stand out using a variety of algorithms. Direct application of fast and greedy method [12] and label propagation algorithms [13] is computationally intensive. Therefore, additional transformations are applied. Image partitions into superpixels [14] are one of these transformations. The superpixel is a small image segment [15]. There are several approaches to breaking an image into superpixels [16–19]. The accuracy of this approach is limited by the superpixels size. Another way to reduce the complexity of community-based segmentation on a graph is based on reducing the amount of information. The threshold principle for determining the graph edge weight reduces the image graph connectivity [20]. This approach segments monochrome images based on highlighting communities on a graph. However, this algorithm is very sensitive to threshold selection. The threshold value is selected manually for each image.

The search for communities in the graph can be used in the post-processing step of the segmented image [21]. The graph is mapped to a segmented image. Some segments are further combined to optimize graph partitioning [21,22]. This approach is effective for images with a large number of small segments [23]. The algorithm for segmentation of the original image based on communities on graphs was proposed in reference [24]. However, this algorithm includes some simplifications. The algorithm considers only the edges for the nearest neighbors. The algorithm has a low speed.

Segmentation based on the graph-theoretic approach and the search for communities can only be applied to a part of the image. The number of pixels decreases significantly after the background is removed. This transformation allows the construction of a fully connected graph based on superpixels [25]. After that, the authors of this work performed clustering of the graph based on geometric features. The third step was to refine the boundaries.

Feature vectors can be used to cluster graphs corresponding to the image [26]. Feature vectors are constructed from the internal structure of graphs and reflect the internal nodes' connectivity. Feature vectors can be used in various graph clustering algorithms.

Algorithms that exist to analyze social networks can be used to detect communities on graphs derived from the image [27,28]. In this case, it is necessary to reduce the number of nodes that are performed using superpixels.

Subgraph splitting methods based on community allocation produce the best results for artificial objects such as cultural heritage artifacts [29]. The eigenvector of the modularity matrix corresponding to the largest positive eigenvalue, the authors recursively identify multilevel subgroups in the graph. The cluster tree strategy creates a cluster map with a variable level of detail. This approach finds the optimal number of clusters.

The main problem with using community search algorithms on graphs for image segmentation is the large amount of computation. One possible solution is to distribute

labels [30]. This approach selects a node label to optimize the objective function by considering the available labels in the neighborhood of the node.

There are several algorithms for detecting the road in an image using a vanishing point: the dominant segment method [31], the parallel line method [32] and the pixel texture orientation method [33]. The most likely area of the road is assessed by the soft voting scheme [33,34] or by comparison with the pattern [35] in these methods. These methods provide a good result for flat asphalt roads with a flat edge. For roads with uneven lighting, the result is poor. Noises and obstacles strongly influence segmentation outcomes. These algorithms have high computational complexity and are not applicable in real-time systems. These algorithms are also not applicable in real road conditions in the presence of cars and pedestrians.

Road selection algorithms based on image segmentation typically involve two steps. In the first step, the algorithm analyzes the color [36,37], texture [38] or road boundaries [39]. The road highlight method can use a combination of these features [40,41]. Machine learning methods highlight the road in the second step [42]. Pre-filtering eliminates most noise. However, large objects cause segmentation errors. Obstacles lead to an incomplete allocation of the road or detect extra areas as part of the road.

Algorithms for marking the road segment based on deep learning of neural networks have been actively used recently [43]. The first convolutional neural networks for road segmentation were end-to-end trainees [44]. Various variants of neural networks solved the same problem [45,46]. Conditional random fields improved segmentation results [47]. Extended convolutions solved the problem of decreasing expansion [48]. The use of artificial neural networks requires human control at the stage of training set formation [49–51]. This approach is the most effective. However, it is expensive and non-scalable. Non-human methods use information about the color and texture of the road [33,52] or a geometric model of the road scene [53].

## 3. Method

We mapped an undirected graph to the image. The graph node corresponds to each pixel of the image. $V$ is the set of the graph nodes. We entered the correlation radius $R$ between the pixels. The correlation radius indicates the maximum distance between pixels whose nodes are connected by an edge in the graph. If $R = 1$ then edges exist only between the nearest neighbors. In this case, the degree of the graph node is 8 if the pixel is not located on the image boundary. If $R = 2$ then each node is connected to the nearest neighbors and the next nearest neighbors. The degree of each node is 24 if the pixel is not located at the image boundary.

The edge weight depends on the color coordinates of the pixels corresponding to the graph nodes. We use the RGB color model to encode the image. The three intensities of the base colors $(r(x, y), g(x, y), b(x, y))$ were mapped to the pixel with coordinates $(x, y)$. $r(x, y)$ is the intensity of red. $g(x, y)$ is the intensity of green. $b(x, y)$ is the intensity of blue. The edge weight was calculated based on the distance between the pixels corresponding to the nodes of the graph.

The five-dimensional space was used to initially represent the image. The point of this space $p = (x, y, r(x, y), g(x, y), b(x, y))$ represents a pixel with coordinates $(x, y)$. The choice of a formula for calculating the distance $d(p, p')$ between points $p$ and $p\prime$ was determined by setting the segmentation problem. The image segment includes pixels close to each other in color and different in color from the remaining pixels. The function $d(p, p')$ should depend on the color difference of the pixels. The distance between pixels must increase as the color coordinate difference increases. Pixels of the same segment must be localized in the image. The function $d(p, p')$ should increase as the geometric distance between pixels increases.

We used the function of the distance between pixels $p = (x, y, r, g, b)$ and $p\prime = (x\prime, y\prime, r\prime, g\prime, b\prime)$ decreasing in exponential law.

$$d(p, p\prime) = \sqrt{(x - x')^2 + (y - y')^2} \left( exp\left( \sqrt{(r - r')^2 + (g - g')^2 + (b - b')^2} \right) - 1 \right). \quad (1)$$

The distance between pixels is zero if their colors match. The distance between pixels grows rapidly as the color coordinate difference increases.

There are other metrics for measuring the distance between points. The Euclidean metric in five-dimensional space requires additional scaling over heterogeneous coordinates.

$$d(p, p') = \sqrt{(x - x')^2 + (y - y')^2 + (r - r')^2 + (g - g')^2 + (b - b')^2}. \quad (2)$$

This metric does not increase quickly enough when the color changes. A metric growing exponentially across all coordinates leads to the dominance of spatial coordinates over color coordinates.

$$d(p, p\prime) = exp\left( \sqrt{(x - x')^2 + (y - y')^2 + (r - r')^2 + (g - g')^2 + (b - b')^2} \right) - 1. \quad (3)$$

Image segmentation is equivalent to cutting a graph into subgraphs in this model. Pixels of one segment must be close to each other in color characteristics and differ from another segment pixels. The formula for edge weight depends on the distance between pixels.

$$w(p, p') = \frac{1}{1 + d(p, p')}. \quad (4)$$

The edge weight is 1 if the pixel colors match. The edge weight decreases as the distance between the color coordinates increases. If the color coordinates are different, then the edge weight decreases as the difference between the geometric coordinates of the pixels increases. Nodes of one segment are strongly connected to each other and weakly connected to nodes outside the segment. Communities on graphs meet these requirements [19]. The segmentation problem boils down to the problem of finding a community on graphs.

We entered the matrix of edge weights for the graph $E$. The element of the matrix $E_{ij}$ is equal to the edge weight between nodes $v_i$ and $v_j$. The matrix $E$ is symmetric with respect to the main diagonal. If the image is $N \times M$, then the matrix size is $NM \times NM$. The matrix is very large for conventional images. A large number of elements in matrix $E$ is zero. The number of non-zero elements depends on the correlation radius $R$. If $R = 1$ then each row has no more than 8 non-zero elements. If $R = 2$ then the number of non-zero elements are not more than 24.

The area growing algorithm was used to search for communities on a graph in this work. The task was to find a community $H$ that includes a node $v \in V$. This process corresponds to finding a segment that includes this point. The average edge weight $W$ of subgraph $H$ is calculated at each step. Nodes from the set $V \setminus H$ associated with at least one node from $H$ were considered at each iteration. The association of such a node with the community nodes $H$ was calculated as the sum of the edge weights.

$$w_i = \sum_{v_j \in H} e_{ij}. \quad (5)$$

If a node's association with a community exceeds a threshold, the node joins the community $H$. The threshold value was calculated based on the average community edge weights $W$. The condition of joining node $v_i$ to the community is written as an inequality.

$$w_i > hW. \quad (6)$$

The parameter $h$ varies the image segmentation accuracy. This parameter determines the color difference that the algorithm perceives as close colors. The correlation radius $R$ also has an effect on segmentation accuracy. This parameter determines the number of edges coming from each node and increases the weight of the connection between the vertex and the community.

The algorithm is recursive. All points of the already formed community were considered at each step. All out-of-community points were checked for each community point. Asymptotic complexity for this algorithm is quadratic in number of pixels $O(M^2 N^2)$.

The average edge weight changes as the community grows. The usual behavior of average community edge weight is presented in Figure 1.

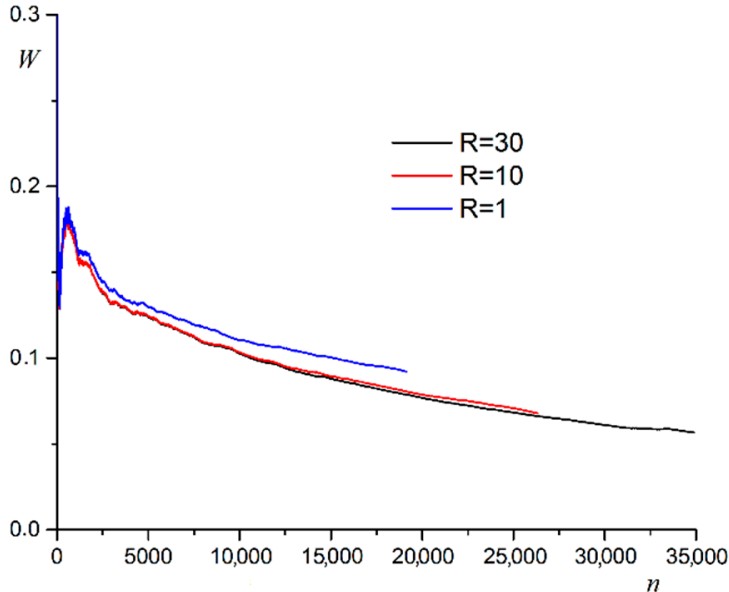

**Figure 1.** Dependence of average edge weight in the community on the number of nodes included at different $R$.

The average edge weight in the community decreases rapidly at first. Subsequently, the average edge weight comes to an asymptotic value. Increasing the correlation radius increases the size of the community. This fact is explained by the increase in the number of neighbors at each node. An increase in the correlation radius $R$ results in a slight decrease in the average edge weight in the community.

The image segment is easily computed from the set of community subgraph nodes. If the image is segmented completely, then segments formed around different points are calculated.

## 4. Computer Experiment

A computer experiment tests an algorithm on color images with a color depth of 1 byte per pixel. We used a computer with a quad-core processor and a frequency of 2600 MHz to carry out the experiment. The software package was implemented in C++. The algorithm highlights a segment in the center of the image. The result was compared to a manually selected segment. The dependence of the results for the algorithm on its parameters was investigated in a computer experiment. The experiment was carried out on a collection of 100 images. The most accurate segmentation was performed at parameter values $h \in [0.005, 0.01]$. An example of algorithm operation at different values of parameter $h$ and $R = 1$ is shown in Figure 2.

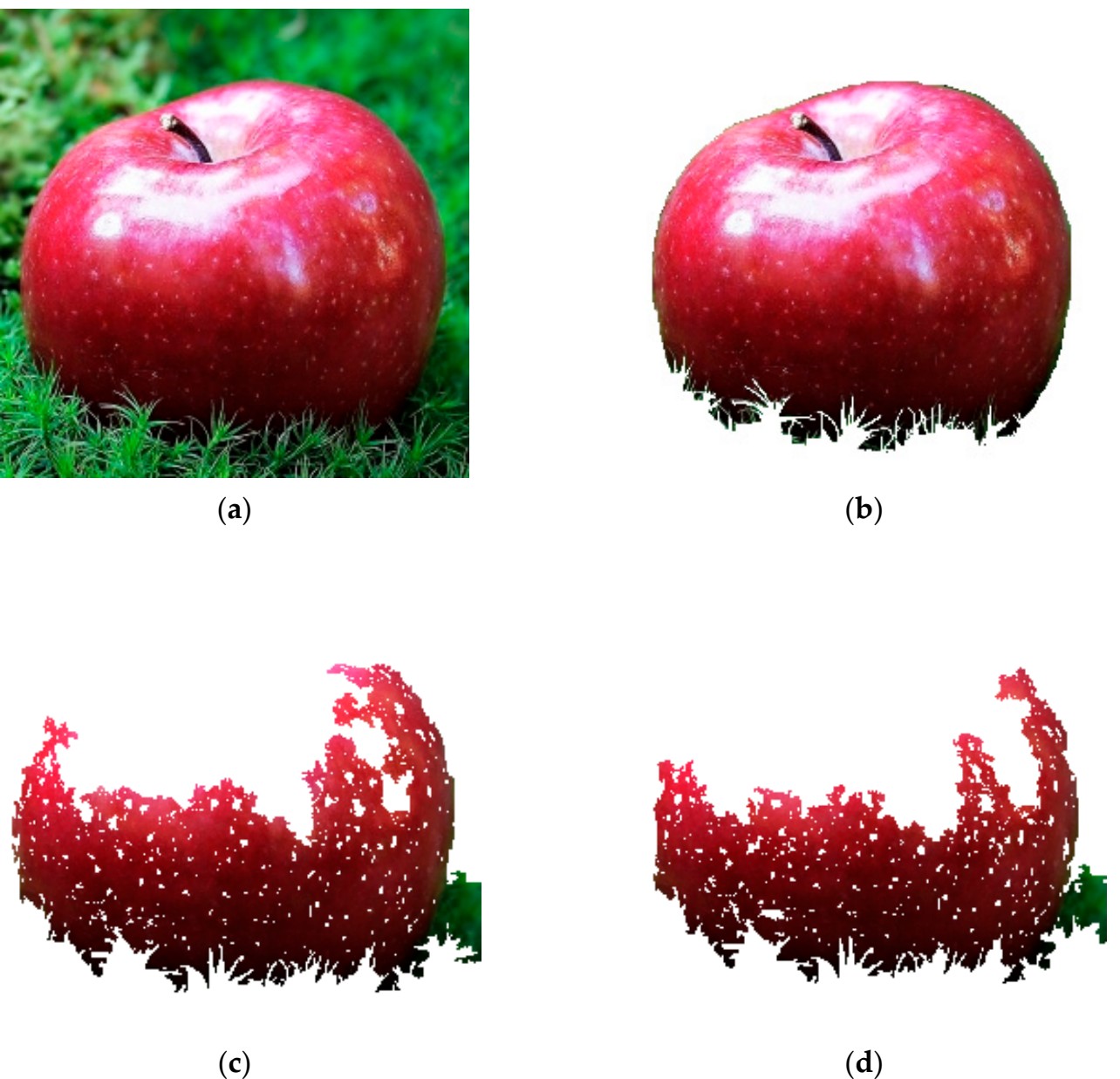

**Figure 2.** Example of algorithm operation at R = 1 and different values of parameter h: (**a**) initial image, (**b**) ground truth, (**c**) h = 0.005, (**d**) h = 0.01.

Increasing the *h* parameter reduces the size of the segment. Two parameters were used to compare segmentation results numerically. The segmentation efficiency is equal to the relative number of pixels correctly assigned by the algorithm to the segment.

$$Eff = \frac{N_t}{N_0} \cdot 100\%. \tag{7}$$

$N_t$ is the number of pixels that the algorithm correctly classified. $N_0$ is the number of pixels in ground truth. The algorithm error is calculated as the ratio of the misclassified pixels number $N_e$ to the total number of pixels in the segment defined by the algorithm $N_s$.

$$Err = \frac{N_e}{N_s} \cdot 100\%. \tag{8}$$

The calculation of these coefficients was performed on the image collection. An average value was calculated for these coefficients. At $R = 1$ and $h = 0.01$ the coefficients are $Eff = 47\%$, $Err = 4.1\%$. At $R = 1$ and $h = 0.001$ the coefficients are $Eff = 56\%$, $Err = 3.8\%$. Lowering the threshold parameter $h$ improves the segmentation accuracy with a slight reduction in errors.

The segmentation accuracy is improved by increasing the correlation radius $R$. Examples of segmentation at $h = 0.007$ and different values $R$ are given in Figure 3.

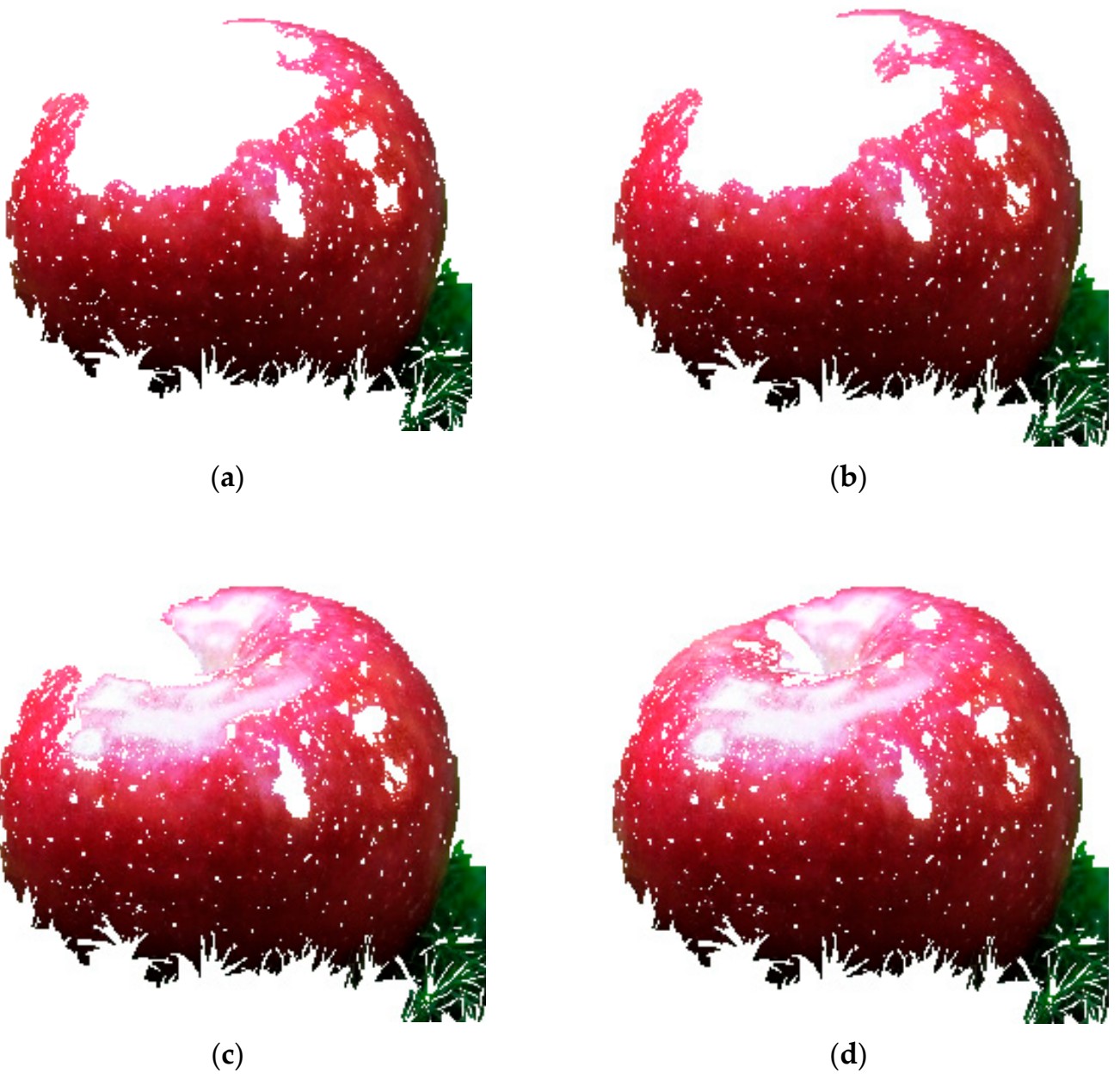

**(a)**          **(b)**

**(c)**          **(d)**

**Figure 3.** Examples of segmentation at $h = 0.007$ and different values $R$: (**a**) $R = 10$, (**b**) $R = 20$, (**c**) $R = 25$, (**d**) $R = 30$.

Increasing the correlation radius $R$ significantly improves segmentation efficiency ($R = 10$ $Eff = 67\%$, $R = 20$ $Eff = 69\%$, $R = 25$ $Eff = 83\%$, $R = 30$ $Eff = 87\%$). Further, increasing the correlation radius does not change the segmentation efficiency. Relative segmentation error decreases as correlation radius increases ($R = 10$ $Err = 8\%$, $R = 20$ $Err = 7\%$, $R = 25$ $Err = 6.7\%$, $R = 30$ $Err = 6\%$). The number of misclassified

nodes is growing slowly. The number of nodes in the community increases rapidly. The ratio of these values decreases.

Examples of segment highlighting in different images are shown in Figure 4.

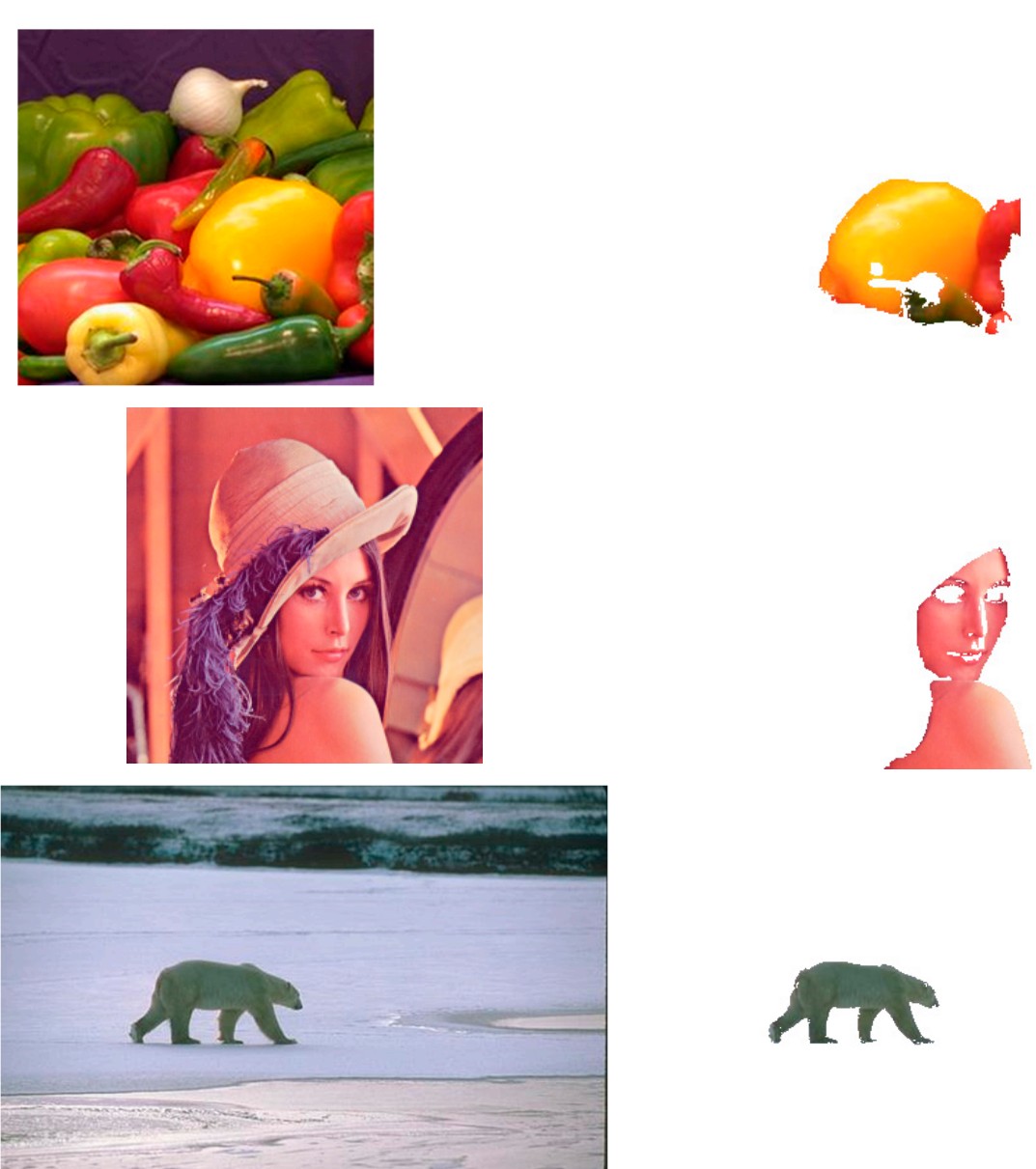

(**a**) (**b**)

**Figure 4.** Examples of segment highlighting in different images: (**a**) source image, (**b**) the result of the algorithm.

## 5. Applying to a Road Image

We used our segmentation algorithm to determine the width of the road from its photo. This problem is relevant for automatic road inventory systems. The camera was mounted on a mobile laboratory. The camera photographed the road through the windshield. The problem is determining the width of the road from this photo of it. The camera parameters and its placement were known.

Calculating the width of a road involves five steps. The road segment is highlighted in the first phase. The boundary of the road segment is defined in the second step. The left and right edge pixels of the road are determined in the third step. The road boundaries are

approximated by straight lines in the fourth step. The width of the road is calculated from the line equations in the fifth step.

Our algorithm was used to highlight a segment of the road. The correct calculation of the segment depends on the selection of the start point. We used an approximate model to image the road in the photo. The image includes four areas: (1) road, (2) sky, (3) objects to the left of the road, and (4) objects to the right of the road. The general model of the road image is shown in Figure 5a. We considered the case of a direct road section outside intersections. This assumption was justified. The camera operator independently selected the shooting moment. The road can go up or down the slope. The position of the chamber can be shifted from the center to the edge of the road. These factors resulted in the displacement of points A, B, C and D in the road image (Figure 5b).

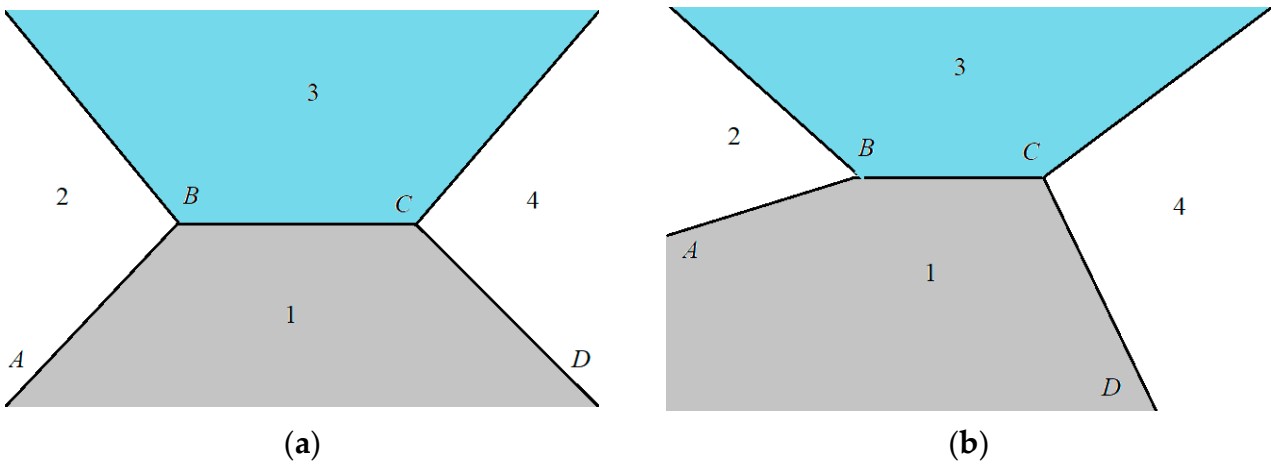

**(a)**　　　　　　　　　　　　　　　　**(b)**

**Figure 5.** Schematic model of the road image: (**a**) direct image of the road, (**b**) image of the road with displacement.

Our algorithm highlights segment 1. The image dimensions are $N \times M$. The point with coordinates $(N/2, M)$ belongs to the road segment. The coordinates of the point were calculated from the upper left corner. However, this is a bad candidate for the starting point of the segment based on the computer experiment results. Road markings are present on real roads. This point is often located on the road marking line. We used the starting point with coordinates $(0.53N, 0.8M)$. Upward displacement is necessary to ignore the image of the windshield bottom. Solid dividing lines are an insurmountable obstacle to the algorithm. Only one lane was highlighted by the algorithm in this case. The algorithm calculated two segments in this case. The starting point of the second segment has coordinates $(0.47, 0.8M)$. The road segment is equal to the combination of the two segments. The two segments match when there is no solid dividing line. The option to manually select an additional segment start point was provided for complex road images with a large number of layout lines.

The segment boundary is defined in the second step. The border has a thickness of one pixel. Boundary pixels are located in the segment. The pixel neighboring the top is located outside the segment. This condition uniquely defines boundary pixels. Examples of highlighting a road segment and its boundaries are shown in Figure 6.

The road edge pixels were highlighted in the third step. The segment boundary was not a straight line for several reasons. Interference includes objects at the edge of the road, the presence of cars and other objects on the road, such as lighting features. The algorithm highlights a section of the minimally distorted edge of the road. Points A and D were calculated as the leftmost and rightmost points of the road outline. Contour areas in a width $0.2N$ were treated as images of the edge of the road. Examples of road outline and road edge images are shown in Figure 7.

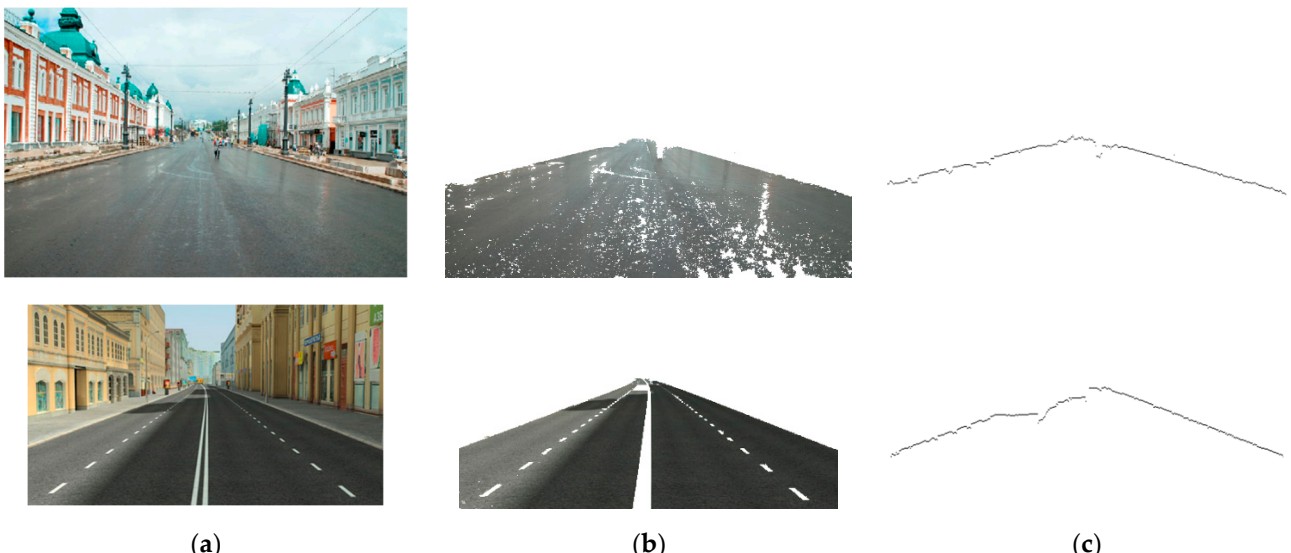

**Figure 6.** Examples of highlighting a road segment (**b**) and its boundaries (**c**), (**a**) is original image.

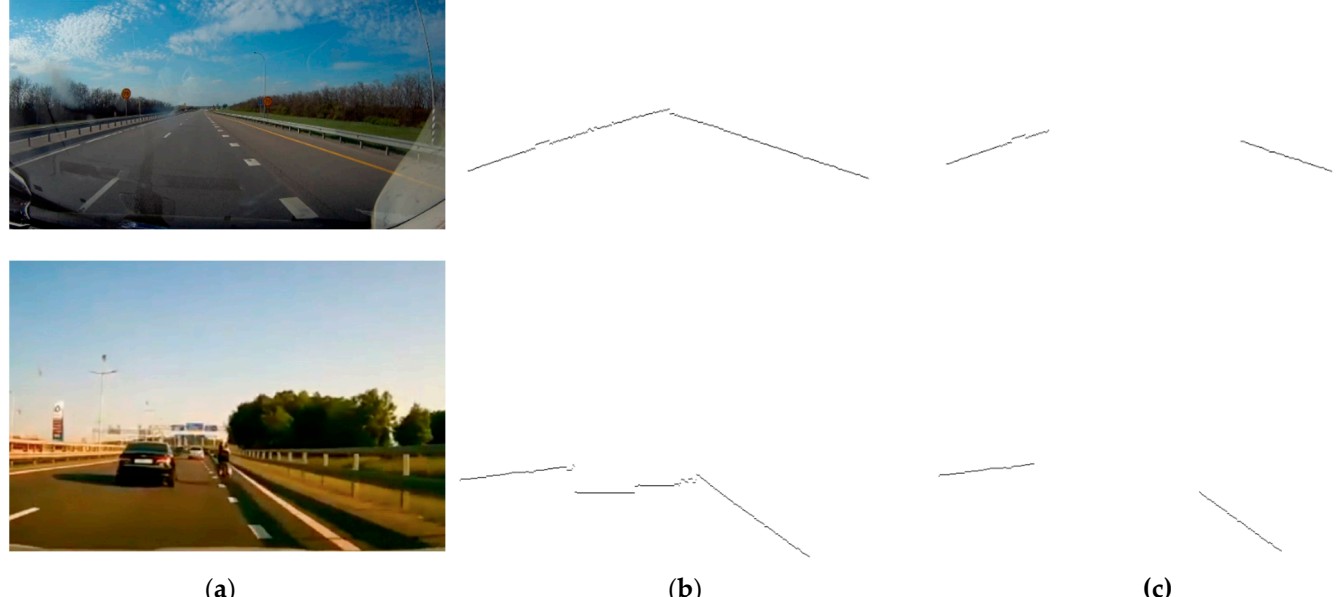

**Figure 7.** Examples of highlighting road boundaries: (**a**) original image, (**b**) road contours, (**c**) road boundaries.

The edges of the road are approximated by straight lines.

$$y = ax + b. \tag{9}$$

The least squares method is used to calculate coefficients $a$ and $b$. The quadratic error function $R$ is calculated for road edge points.

$$R = \sum_{i=1}^{N} (ax_i + b - y_i)^2. \tag{10}$$

$N$ is the number of dots in the one edge road image. $(x_i, y_i)$ are coordinates of points at the road edge. Coefficients a and b are calculated based on the condition that the quadratic error function $R$ is minimal.

$$\frac{\partial R}{\partial a} = 0, \quad \frac{\partial R}{\partial b} = 0. \tag{11}$$

The solution of this equations system is expressed through the coordinates of the road edge points.

$$a = \left( \sum_{i=1}^{N} y_i \cdot \sum_{i=1}^{N} x_i - N \cdot \sum_{i=1}^{N} x_i y_i \right) \Big/ \left( \left( \sum_{i=1}^{N} x_i \right)^2 - N \cdot \sum_{i=1}^{N} x_i^2 \right), \tag{12}$$

$$b = \left( \sum_{i=1}^{N} x_i y_i \cdot \sum_{i=1}^{N} x_i - \sum_{i=1}^{N} y_i \cdot \sum_{i=1}^{N} x_i^2 \right) \Big/ \left( \left( \sum_{i=1}^{N} x_i \right)^2 - N \cdot \sum_{i=1}^{N} x_i^2 \right). \tag{13}$$

The coefficients were calculated independently for the left and right road edges. The result is two pairs of coefficients $(a_1, b_1)$ and $(a_2, b_2)$. The first coefficients $(a_1, b_1)$ correspond to the straight left edge of the road. The second coefficients $(a_2, b_2)$ determine the right edge of the road. These two lines intersect at point $(x_0, y_0)$.

$$x_0 = \frac{b_2 - b_1}{a_1 - a_2}, \quad y_0 = \frac{a_1 b_2 - a_2 b_1}{a_1 - a_2}. \tag{14}$$

Examples of road edge approximation for the road segment and the original image are shown in Figure 8. The road edge lines are drawn to the intersection point.

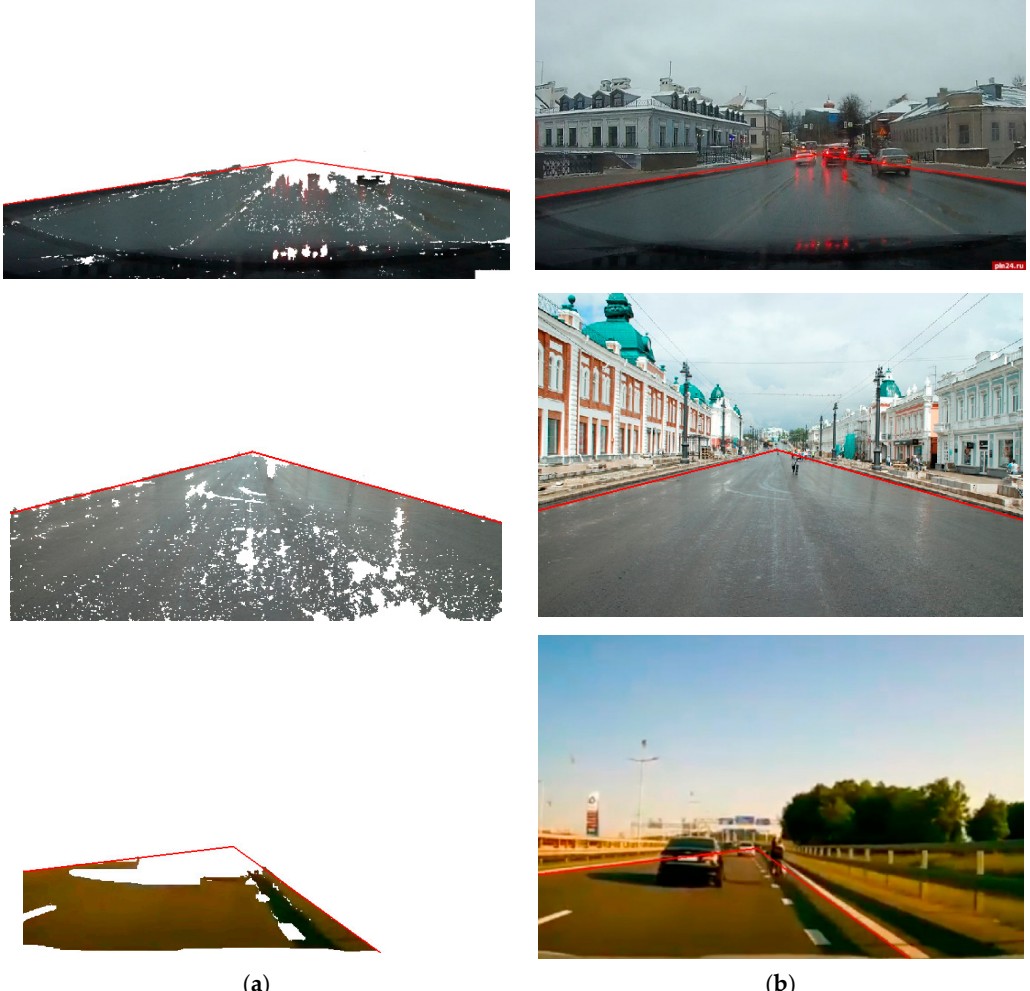

(**a**)　　　　　(**b**)

**Figure 8.** Examples of road edge approximation for the road segment (**a**) and the original image (**b**). Red lines show the border of the road.

### 6. Calculating the Width of a Road

We considered rays in the optical system to determine the real width of the road. Parameter $D$ is the real width of the road. Parameter $d$ is the width of the road in the image. The width of the road in the image depends on the measurement point due to the perspective effect. Parameter $y_1$ is the height from the top edge of the image at which the road width $d$ is measured. This point in the image corresponds to a point at a distance $S$ from the camera. Parameter $h$ is the height of the camera above the road. Parameter $r$ is the distance from the lens to the matrix in the camera. The optical system parameters are shown in Figure 9.

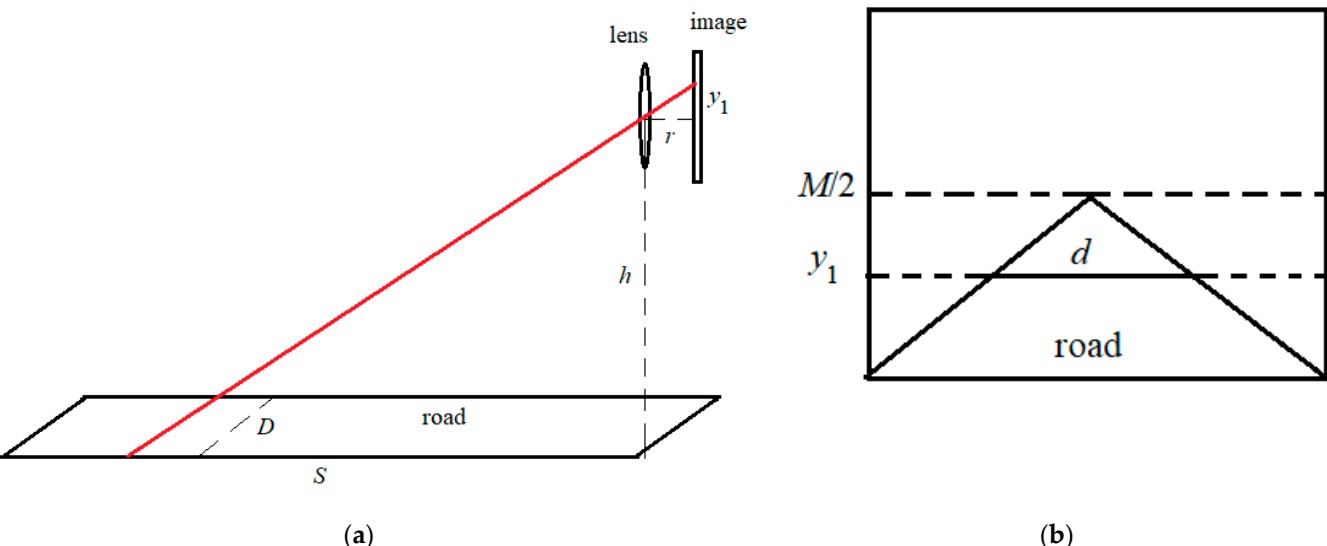

(a)                                                                 (b)

**Figure 9.** The optical system parameters. (**a**) Imaging the road in the lens, (**b**) parameters in the image.

The height of the point in the image and the distance to the point on the road are related by the formula (Figure 9a).

$$\frac{h}{S} = \frac{y_1}{r}. \tag{15}$$

We write down the formula for a thin lens.

$$\frac{1}{S} + \frac{1}{r} = \frac{1}{f}. \tag{16}$$

Parameter $f$ is the focal length of the lens.

$$r = \frac{fS}{S - f}. \tag{17}$$

This expression is substituted in formula (13).

$$S = f\left(1 + \frac{h}{y_1}\right). \tag{18}$$

The selection of fixed values $h$ and $y_1$ results in measuring the width of the road at the same distance from the camera. If the shooting is performed on a fixed hardware platform, then $h$ remains the same. Value $y_1$ is a parameter of the algorithm. This parameter is fixed in the program.

We write down the formula for the width of the road (Figure 9b).

$$\frac{D}{S} = \frac{d}{r}. \tag{19}$$

The expression (15) is substituted into this formula.

$$D = d\frac{S - f}{f}.$$ (20)

The formula (16) is used for $S$.

$$D = d\frac{h}{y_0}.$$ (21)

$d$ is measured in meters in this formula. The algorithm measures $d$ in pixels in the image. The coefficient $t$ converts the distance in pixels into meters. $t$ depends on the resolution of the camera.

$$D = d\frac{th}{y_1}.$$ (22)

The new factor $k$ is introduced for ease of calculation.

$$k = th.$$ (23)

This factor defines the specifications for image acquisition. It depends on the camera used and its position.

$$D = d\frac{k}{y_1}.$$ (24)

Parameter $k$ can be obtained by calibration the system. Several photos of roads with a known width are necessary for calibration. The algorithm determines the width $d$ for these shots. Coefficient k is calculated from known values $D$, $d$ and $y_1$. The average value $k$ is used in further system operation.

The width of the road in the photo is calculated based on the straight equations for the road boundaries. The value $y_1$ is substituted in Equation (9).

$$\begin{aligned} y_1 &= a_1 x_1 + b_1, \\ y_1 &= a_2 x_2 + b_2. \end{aligned}$$ (25)

The coordinate values $x_1$ and $x_2$ are calculated from these equations. The width of the road is defined as the difference between the two values.

$$d = x_2 - x_1.$$ (26)

The width of the road in the image is expressed through the equation's coefficients for lines.

$$d = y_1\left(\frac{1}{a_2} - \frac{1}{a_1}\right) + \left(\frac{b_1}{a_1} - \frac{b_2}{a_2}\right).$$ (27)

The width of the real road is calculated by substituting formula (27) into formula (25).

$$D = k\left(\frac{1}{a_2} - \frac{1}{a_1} + y_1\left(\frac{b_1}{a_1} - \frac{b_2}{a_2}\right)\right).$$ (28)

We used a value of $y_1 = 3M/4$. $M$ is the height of the image in pixels.

These formulas apply only to the horizontal road. If the road is on the rise, then the formula gives an increased value for the width of the road. If the road slopes, then the formula gives an underestimated value. The presence of a road slope or rise is determined by the position of the horizon line in the image. For a horizontal road, the horizon line is located in the middle of the image. The horizon line is above the middle of the image for rising. The horizon line is below the middle of the slope. The algorithm cannot highlight the horizon line in the image due to the presence of objects obscuring it. One point on the horizon line is calculated as the intersection of the straight road boundaries $(x_0, y_0)$. This point is sufficient to define the entire horizon line. If the road is horizontal, then $y_0 = M/2$.

$y_0$ is greater than $M/2$ for the rising ($y_0 < M/2$). The coordinate $y_0$ is less than $M/2$ for the slope $M/2$ ($y_0 > M/2$).

The geometric transformation of the road image is performed to obtain the true value of the road width at $y_0 \neq M/2$. This transformation is demonstrated in Figure 10.

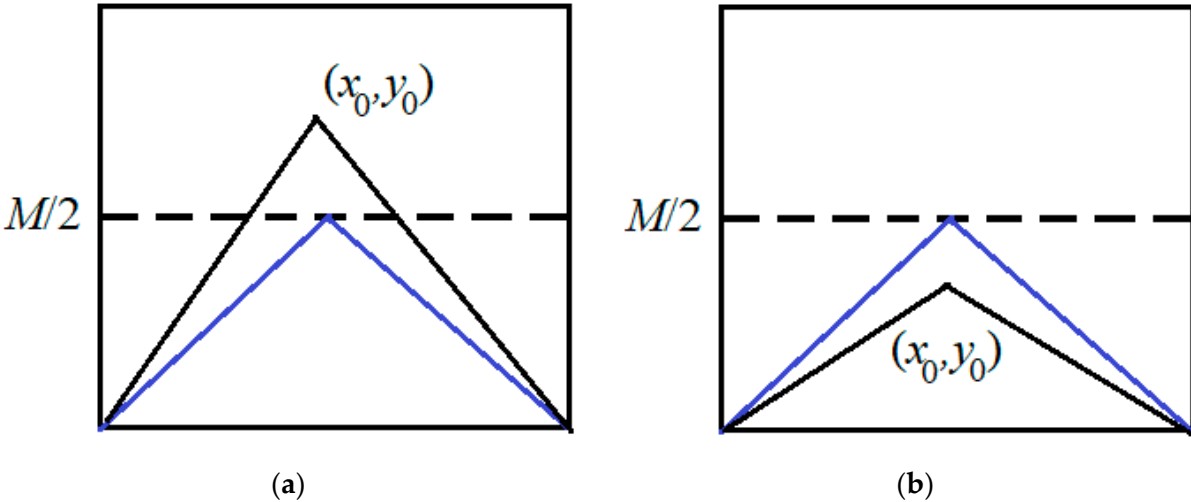

|      |      |
| :--: | :--: |
| (**a**) | (**b**) |

**Figure 10.** The road image is converted at $y_0 \neq M/2$. The black line is the original image of the road. A blue line is an image of a road after conversion ((**a**) rising, (**b**) slope).

The image is compressed along the OY axis at $y_0 < M/2$. The image stretches along the OY axis at $y_0 > M/2$. The equations for straight approximating road edges change after conversion.

$$
\begin{aligned}
y &= a_1 \frac{M}{2y_0} x + b_1 \frac{M}{2y_0}, \\
y &= a_2 \frac{M}{2y_0} x + b_2 \frac{M}{2y_0}.
\end{aligned} \tag{29}
$$

The true value of the road width is calculated using the modified formula.

$$
D = k \frac{2y_0}{M} \left( \frac{1}{a_2} - \frac{1}{a_1} + y_1 \left( \frac{b_1}{a_1} - \frac{b_2}{a_2} \right) \right). \tag{30}
$$

We tested the proposed algorithm on a collection of 100 images. Images were obtained by the same camera mounted on a car. The training set included five road images of known width. The road width of the training set was measured manually. The road segment was highlighted for all images in the collection. Road widths were measured manually to check the accuracy of the algorithm. The difference between the results of the algorithm and the results of manual measurement do not exceed 5%.

## 7. Conclusions

Our segmentation algorithm is based on representing the image as a weighted graph. Image segmentation is performed as a selection of subgraphs whose nodes are more strongly connected to each other than to other nodes. Our algorithm differs in two ways from other algorithms. Edges are connected, not only nodes of the nearest neighboring pixels, but also nodes at some distance from each other in graph construction. Considering long-range correlations increases the efficiency of image segmentation. Increasing the radius of node correlation increases the running time of the algorithm. There is an optimal correlation radius $R = 30$. The efficiency of the algorithm remains unchanged when this threshold is exceeded. The second feature of the algorithm is to consider not the weight of the edge with the nearest community node, but the total weight of all edges within the correlation radius. This approach allows the algorithm to bypass point features in the image. The communication threshold for including a node in a community is constantly

changed and calculated based on the edges of the formed community. This allows the algorithm to segment images with gradient fill. However, this approach leads to an increase in the number of nodes mistakenly included in the community in the absence of clear boundaries in the image.

We compared the results of our algorithm with similar works. Images of 200 different roads were used for testing. The images were obtained using a video recorder and from various sources on the Internet. Figure 11 shows examples of the original images, ground truth, the results of articles [54,55], the results of our algorithm.

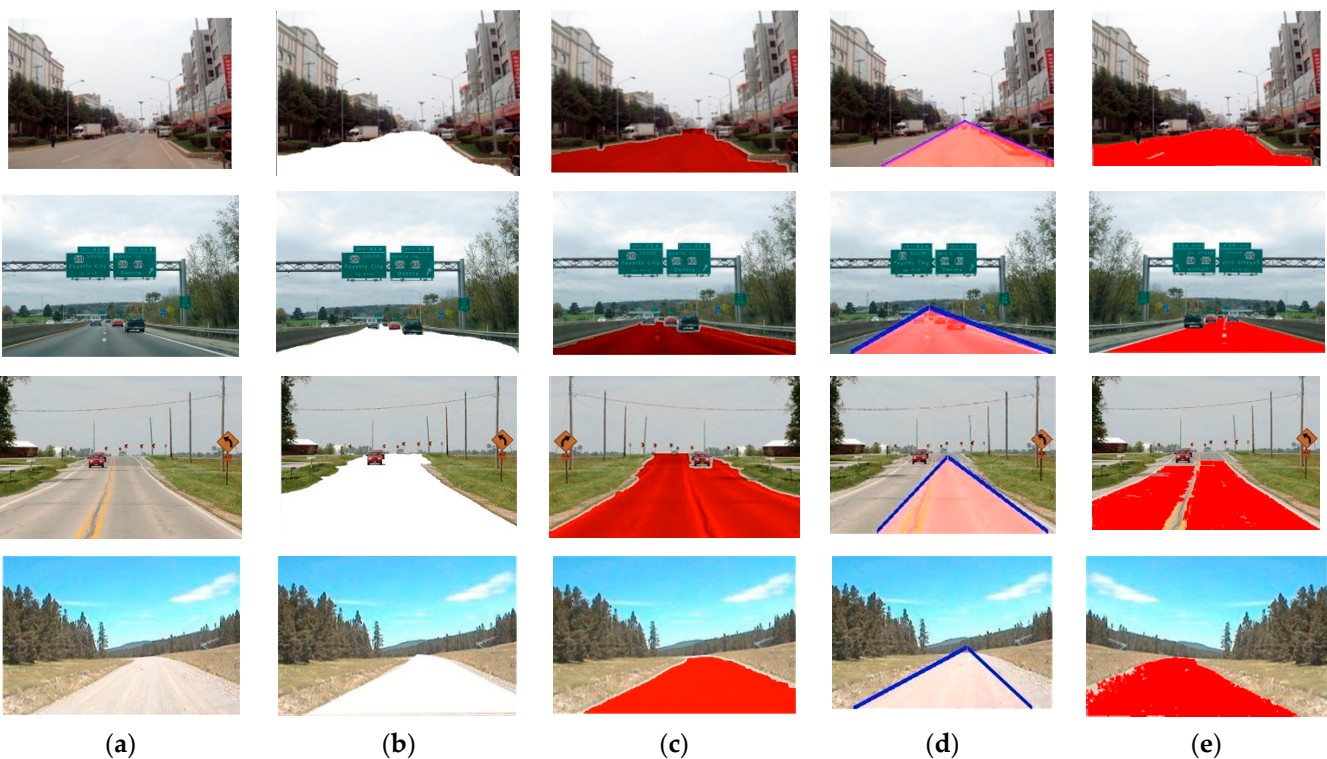

| (a) | (b) | (c) | (d) | (e) |

**Figure 11.** Segmentation examples: (**a**) original images, (**b**) ground truth, (**c**) [54], (**d**) [55], (**e**) our algorithm.

As can be seen from Figure 11, our algorithm provides segmentation results close to ground truth. The algorithm from article [54] produces similar results in terms of segmentation quality. However, our algorithm works faster. All images were 492 × 318 in size. Average operating time of the algorithm in [54] was 6.9 s. The average running time of the algorithm in [55] was 7.1 s. A similar algorithm from article [56] works in 957.2 s. Our algorithm performs segmentation on the same set of images over an average time of 2.3 s.

**Author Contributions:** Conceptualization, S.V.B.; methodology, S.Y.B.; software, S.V.B.; formal analysis, S.Y.B.; writing—review and editing, S.V.B. and S.Y.B. All authors have read and agreed to the published version of the manuscript.

**Funding:** This research received no external funding.

**Informed Consent Statement:** Informed consent was obtained from all subjects involved in the study.

**Conflicts of Interest:** The authors declare no conflict of interest.

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
