# Peer review of "Images Segmentation Based on Cutting the Graph into Communities"

_algorithms, doi:10.3390/a15090312_

Round 1

Reviewer 1 Report

Editorial comment to the title. The title should be printed/composed as follows:

Images segmentation based on cutting the graph

into communities

Motivation for the choice of particular forms of formulae (1) and (2) is needed, i.e. some other possible formulae needs to be presented and their features should be compared to (1) and (2) – if the authors know any.

Line 131: It should be:  . . . elements in matrix E is . . . (italic)  

Lines 149/150: For completeness of presentation the justification for the complexity is needed.

Editorial remark: The figures are out of the text area (placed on the left side of the pages).

Two formulae (9): An empty space between the two formulae is needed.

Lines 267 – 272: The sentences can not begin with symbols like d, h and the like.

Formulas from (15) to (22). Is it possible that S=f ? Is it possible that one of the values y1, S, r, f, y0 is equal zero ?    

Author Response

Dear reviewer!

We thank you for your constructive comments. We provide answers to your questions and comments.

  1. The title should be printed/composed as follows:

«Images segmentation based on cutting the graph into communities»

Answer: We have inserted the title in your proposed revision

  1. Motivation for the choice of particular forms of formulae (1) and (2) is needed, i.e. some other possible formulae needs to be presented and their features should be compared to (1) and (2) – if the authors know any.

Answer: We added a paragraph.

“There are other metrics for measuring the distance between points. The Euclidean metric in five-dimensional space requires additional scaling over heterogeneous coordinates.

This metric does not increase quickly enough when the color changes. A metric growing exponentially across all coordinates leads to the dominance of spatial coordinates over color coordinates.

  1. Line 131: It should be: . . . elements in matrix E is . . . (italic)

Answer: The text has been adjusted.

  1. Lines 149/150: For completeness of presentation the justification for the complexity is needed.

Answer: We added a paragraph.

“The algorithm is recursive. All points of the already formed community are considered at each step. All out-of-community points are checked for each community point. Asymptotic complexity for this algorithm is quadratic in number of pixels .”

  1. Two formulae (9): An empty space between the two formulae is needed.

Answer: The text has been adjusted.

  1. Lines 267 – 272: The sentences can not begin with symbols like d, h and the like.

Answer: The text has been adjusted.

«Параметр — это реальная ширина дороги. Параметр — ширина дороги на изображении. Ширина дороги на изображении зависит от точки измерения из-за эффекта перспективы. Параметр — высота от верхнего края изображения, на которой измеряется ширина дороги. Эта точка на изображении соответствует точке на расстоянии от камеры. Параметр — высота камеры над дорогой. Параметр — это расстояние от объектива до матрицы в камере».

  1. Формулы от (15) до (22). Возможно ли, что S=f ? Возможно ли, чтобы одно из значений y1, S, r, f, y0 было равно нулю?

Ответ: Приведем параметры алгоритма, при которых эти значения не могут быть равны нулю.

Reviewer 2 Report

The authors present an heuristic approach for image segmentation. The manuscript reads well and results are sound. My major concerns are listed as follow: 

1) Novelty: not clear where the novelty resides in the proposed approach;

2) Computational times of the presented algorithm are not specified. it would be interesting to understand whether the method can be applied in real-time scenarios

3) No comparison with state-of-the-art techniques in terms of quantitative metrics. 

Author Response

Dear reviewer!

We thank you for your constructive comments. We provide answers to your questions and comments.

1) Novelty: not clear where the novelty resides in the proposed approach;

Answer: We added information and extended the conclusion.

2) Computational times of the presented algorithm are not specified. it would be interesting to understand whether the method can be applied in real-time scenarios

Answer: We added information and extended the conclusion.

3) No comparison with state-of-the-art techniques in terms of quantitative metrics.

Answer: We added information and extended the conclusion.

Reviewer 3 Report

The presented manuscript describes the segmentation technique by graph-cut and its application for road segmentation. Despite some interesting findings this paper is rather weak in a present form because it is hard to see a novelty and contribution. 

1. A scientific paper have to contain a critical review of previous works on the subject under consideration. There are plenty papers devoted to various graph-cut segmentation algorithms. "Related works" section mentions only 18 publications. It is absolutely insufficient for so wide area as image segmentation via graph-cut. The road segmentation is considered as a practical application of the developed method. Again, there are many publications about road segmentation on a photo. None from existing papers devoted to road segmentation is in References.  In general, 29 items in the list of references look rather small for a journal paper.

2. A novelty and contribution of the proposed method should be explained and described clearly. What is a new in comparison with existing algorithms? What are advantages of proposed technique? Section "Conclusions" tries to demonstrate differences of the proposed algorithm with existing ones, but it is too short and vague. It is necessary to emphasize a difference of your proposal from other approaches in whole text starting from Introduction. 

3. It is not a good style to demonstrate results of own method only; results should be demonstrated for several various algorithms. For example, you can compare results of the proposed algorithm with outcomes of GrabCut from OpenCV or label propagation (https://github.com/benedekrozemberczki/LabelPropagation). 

4. Extensive English editing is necessary. There are many uncommon terms in the text. It is better to say "ground truth" instead of "manual segmentation". It is more correct to write in line 293 about "calibration" instead of "training". There are dozens of similar examples.

Usage of a passive voice should be decreased considerably, especially in an Abstract. 

"Fast Greedy" is not name of the algorithm from [11]. Indeed sometimes that method is mentioned in the publication as fast and greedy (adjectives), but it is not name starting from capital letter. It can be assumed that instead of "Label Propaganda" you meant "label propagation". However, again, it is not name of method from [12]. 

5. Detailed description way for calculation of road width is not related to the main subject of the paper - segmentation. In addition, statements from (7) to (25) look rather obvious.  I recommend to remove it from the revised version of the paper. Other option is targeting of the paper on road width estimation instead of general segmentation technique. 

In general, I'm sure the paper can be revised to do a high-quality publication. 

Author Response

Dear reviewer!

We thank you for your constructive comments. We provide answers to your questions and comments.

  1. A scientific paper have to contain a critical review of previous works on the subject under consideration. There are plenty papers devoted to various graph-cut segmentation algorithms. "Related works" section mentions only 18 publications. It is absolutely insufficient for so wide area as image segmentation via graph-cut. The road segmentation is considered as a practical application of the developed method. Again, there are many publications about road segmentation on a photo. None from existing papers devoted to road segmentation is in References. In general, 29 items in the list of references look rather small for a journal paper.

Answer: The Related Works section provides links to articles about using community highlighting to segment an image. A full overview of segmentation methods based on graph cutting requires writing a separate review paper rather than a section in the research paper. The section " Related Works " has been supplemented with a description of articles on the allocation of a road segment.

  1. A novelty and contribution of the proposed method should be explained and described clearly. What is a new in comparison with existing algorithms? What are advantages of proposed technique? Section "Conclusions" tries to demonstrate differences of the proposed algorithm with existing ones, but it is too short and vague. It is necessary to emphasize a difference of your proposal from other approaches in whole text starting from Introduction.

Answer:

We added a paragraph in the introduction.

“Algorithms for highlighting communities on graphs provide a high level of image segmentation quality. However, these algorithms have high computational complexity and long running times. These disadvantages do not allow these algorithms to be used directly. Community highlighting algorithms work in the image preprocessing or image postprocessing stages. The purpose of the paper is to develop an image segmentation algorithm based on highlighting communities on a graph with acceptable running time. The developed algorithm is used to highlight the road segment.”

We added information and extended the conclusion.

  1. It is not a good style to demonstrate results of own method only; results should be demonstrated for several various algorithms. For example, you can compare results of the proposed algorithm with outcomes of GrabCut from OpenCV or label propagation (https://github.com/benedekrozemberczki/LabelPropagation).

Answer: We added information and extended the conclusion.

  1. Extensive English editing is necessary. There are many uncommon terms in the text. It is better to say "ground truth" instead of "manual segmentation". It is more correct to write in line 293 about "calibration" instead of "training". There are dozens of similar examples.

Answer: We have corrected the terms.

Usage of a passive voice should be decreased considerably, especially in an Abstract.

Answer: We have corrected the text and the Abstract.

"Fast Greedy" is not name of the algorithm from [11]. Indeed sometimes that method is mentioned in the publication as fast and greedy (adjectives), but it is not name starting from capital letter. It can be assumed that instead of "Label Propaganda" you meant "label propagation". However, again, it is not name of method from [12].

Answer: We have corrected the terms.

  1. Detailed description way for calculation of road width is not related to the main subject of the paper - segmentation. In addition, statements from (7) to (25) look rather obvious. I recommend to remove it from the revised version of the paper. Other option is targeting of the paper on road width estimation instead of general segmentation technique.

Answer:

The road width calculation section justifies the purpose of solving the problem of selecting a road segment in the image. This section shows the application value of the segmentation problem being solved. Removing this section makes the segmentation problem abstract and less interesting. The document aims to segment the image as the most difficult part of a complex problem. Formulas (7) to (15) may appear obvious. However, deleting them makes the article less readable and less understandable. Therefore, we believe that the preservation of these formulas expands the audience of readers who have access to our article.

Round 2

Reviewer 2 Report

The manuscript greatly improved after revision. The revised version is clear and more focused. 

One very minor comment: 

- From Figure 7 it appears that some sub-figures in captions (c) and (e) are flipped along the Y axis, why? 

Author Response

We changed the caption under Figure 7.

«Examples of highlighting road boundaries: (a) original image, (b) road contours, (c) road boundaries.»

Reviewer 3 Report

The presented manuscript looks much better after revision. Nevertheless some additional improvements are absolutely necessary.

1. The results presentation:

a. The outcomes of road segmentation should be moved to separate section located before Conclusion. Despite qualitative results in figure 7 demonstrate advantages of the proposed method over [55] and [56], an addition of numerical results are prompted as well. It is easy to calculate Eff (7) and Err (8) metrics due to ground truth is available. It would be nice to form numerical results in the form of tables or diagrams, not just text. Additional details about the testing dataset are desirable. How many images contain? (Is it really just five images?) Where was it collected? Is it public or not? Etc.

b. A reader need to know exact information about computing platform (both hardware and software) to be able understand a computational complexity and/or efficiency of the techniques under comparison. An indication only processing time is not enough. What were used computer, OS, programming language for the computing experiments?

c.    It is recommended to present results from Section 4 as a table or diagram/plot, not only text. 

2. Used terms in English: The text uses a lot of terms and even technical slang that may be incomprehensible outside the Russian-speaking community. The issue was pointed out in my previous review, however only directly indicated points were corrected in the revised version.  A use of uncommon terminology and unclear statements is shortcoming of presented manuscript. It is recommended for authors to borrow terminology from the modern publications about image segmentation prepared by famous researchers.

Just for example, because dozens of similar cases can be found in the text:

a. Let's consider Eff (7) and Err (8). Those are metrics or measures or Figure of Merit (FoM), but it is not coefficients or parameters. "Nt is the number of pixels that the algorithm correctly classified". That is Nt is the number of true positives (TP). "No is the number of pixels in ground truth." Is it total number of pixels, that is an image size?  Assumably, authors meant the number of pixels of positive segment in ground truth.  In that case No = TP + FN; Eff = TP/(TP + FN); and Eff is the same as Recall (or sensitivity, or hit rate, or true positive rate). "Ne is the misclassified pixels number". Is it Ne =FP + FN? "Ns the total number of pixels in the segment defined by the algorithm". Is Ns = TP + FP? Why to invent own performance metrics instead of commonly used? 

b. "A computer experiment tests the algorithm on color images with a color depth of 256." Color depth or bit depth is in bits per pixel. What is 256?

Author Response

We provide answers to your questions and comments.

  1. The results presentation:

  1. The outcomes of road segmentation should be moved to separate section located before Conclusion. Despite qualitative results in figure 7 demonstrate advantages of the proposed method over [55] and [56], an addition of numerical results are prompted as well. It is easy to calculate Eff (7) and Err (8) metrics due to ground truth is available. It would be nice to form numerical results in the form of tables or diagrams, not just text. Additional details about the testing dataset are desirable. How many images contain? (Is it really just five images?) Where was it collected? Is it public or not? Etc.

Answer: The metrics Eff and Err have not changed. Why list them a second time?

We've added text about a set of test images.

«Images of 200 different roads were used for testing. The images were obtained using a video recorder and from various sources on the Internet.»

We do not believe that adding tables or diagrams will improve the perception of the results.

  1. A reader need to know exact information about computing platform (both hardware and software) to be able understand a computational complexity and/or efficiency of the techniques under comparison. An indication only processing time is not enough. What were used computer, OS, programming language for the computing experiments?

Answer: We added the following phrase:

«We used a computer with a quad-core processor and a frequency of 2600 MHz to carry out the experiment. The software package is implemented in C++.»

Specifying a computer and operating system manufacturer is an advertisement. The international scientific bases SCOPUS and Web of Science have a negative attitude towards advertising in the text of scientific articles and do not index such articles!

  1. It is recommended to present results from Section 4 as a table or diagram/plot, not only text.

Answer: We do not consider the presentation of this data as a table to be informative or useful. The table will clutter the article. The reader will have to deal with the structure of the table. Explanations to the table will take up a lot of space and make the article less readable.

  1. Used terms in English: The text uses a lot of terms and even technical slang that may be incomprehensible outside the Russian-speaking community. The issue was pointed out in my previous review, however only directly indicated points were corrected in the revised version. A use of uncommon terminology and unclear statements is shortcoming of presented manuscript. It is recommended for authors to borrow terminology from the modern publications about image segmentation prepared by famous researchers.

Answer: We perceive this remark as incorrect and nationalist. You argue that the Russian-speaking community is not like the global one! We recommend that you find another platform for your nationalist activities! This remark is contrary to scientific ethics!

Just for example, because dozens of similar cases can be found in the text:

  1. Let's consider Eff (7) and Err (8). Those are metrics or measures or Figure of Merit (FoM), but it is not coefficients or parameters. "Nt is the number of pixels that the algorithm correctly classified". That is Nt is the number of true positives (TP). "No is the number of pixels in ground truth." Is it total number of pixels, that is an image size? Assumably, authors meant the number of pixels of positive segment in ground truth. In that case No = TP + FN; Eff = TP/(TP + FN); and Eff is the same as Recall (or sensitivity, or hit rate, or true positive rate). "Ne is the misclassified pixels number". Is it Ne =FP + FN? "Ns the total number of pixels in the segment defined by the algorithm". Is Ns = TP + FP? Why to invent own performance metrics instead of commonly used?

Answer: You suggest using four values instead two values. This replacement complicates the formulas. Designations and terms are not dogma. It is necessary to use such designations that are convenient in a specific situation. Our designations are less common than those suggested by you, but are common.

  1. "A computer experiment tests the algorithm on color images with a color depth of 256." Color depth or bit depth is in bits per pixel. What is 256?

Answer: The new revision has the form:

«A computer experiment tests an algorithm on color images with a color depth of 1 byte per pixel.»